# Pollution exacerbates China's water scarcity and its regional inequality

Ting Ma [1,2,3,8]*, Siao Sun [4,8], Guangtao Fu [5], Jim W. Hall [6], Yong Ni [1,2,7]*, Lihuan He [7], Jiawei Yi [1,2], Na Zhao[1,2], Yunyan Du[1,2], Tao Pei [1,2,3], Weiming Cheng[1,2], Ci Song [1,2], Chuanglin Fang[2,4] & Chenghu Zhou [1,2]*

Inadequate water quality can mean that water is unsuitable for a variety of human uses, thus exacerbating freshwater scarcity. Previous large-scale water scarcity assessments mostly focused on the availability of sufficient freshwater quantity for providing supplies, but neglected the quality constraints on water usability. Here we report a comprehensive nationwide water scarcity assessment in China, which explicitly includes quality requirements for human water uses. We highlight the necessity of incorporating water scarcity assessment at multiple temporal and geographic scales. Our results show that inadequate water quality exacerbates China's water scarcity, which is unevenly distributed across the country. North China often suffers water scarcity throughout the year, whereas South China, despite sufficient quantities, experiences seasonal water scarcity due to inadequate quality. Over half of the population are affected by water scarcity, pointing to an urgent need for improving freshwater quantity and quality management to cope with water scarcity.

[1] State Key Laboratory of Resources and Environmental Information System, Institute of Geographical Sciences and Natural Resources Research, Chinese Academy of Sciences, Beijing 100101, China. [2] University of Chinese Academy of Sciences, Beijing 100049, China. [3] Jiangsu Center for Collaborative Innovation in Geographical Information Resource Development and Application, Nanjing 210023, China. [4] Key Laboratory of Regional Sustainable Development Modeling, Institute of Geographical Sciences and Natural Resources Research, Chinese Academy of Sciences, Beijing 100101, China. [5] Centre for Water Systems, University of Exeter, Exeter EX4 4QF, UK. [6] Environmental Change Institute, University of Oxford, Oxford, UK. [7] China National Environmental Monitoring Center, Beijing 100012, China. [8] These authors contribute equally: Ting Ma, Siao Sun *email: mting@lreis.ac.cn; niyong@cnemc.cn; zhouch@lreis.ac.cn

The survival and development of human society depends on water, and the global demand has increased by nearly eightfold over the period 1900–2010[1,2], driven by population growth, expanding irrigated croplands, economic development, and dietary shifts[2–8]. Increasing water demands in combination with their geographic and temporal mismatch with freshwater availability have rendered water scarcity a widespread problem in many parts of the world, which occurs when demand for freshwater exceeds available supply[9,10]. Particularly in China, with per capita available water resource amounting to only one fourth of the world average[11–13], water scarcity is one of the most significant threats that challenge sustainable development[14–16]. The uneven distributions of water and population create inequality in water scarcity[13], with some regions in North China facing extreme water pressures to an extent that is not revealed by national average figures. As a result of rising conflicts among regional and sectoral water uses, policy attention on mitigating water scarcity is growing in China[16–18].

Understanding water scarcity should underpin sustainable water resources management[10,19,20]. Many previous studies have assessed China's water scarcity at regional and national levels[21–24], or in a global context[4,25–27]. Previous water scarcity assessments mostly focused on the quantity of water available for water supplies[3,26,28–32], but neglected the fact that inadequate water quality may pose a significant constraint on water usability[33,34]. The dramatic economic development in recent decades in China has come at an environmental cost, where widespread land use changes, increasing volumes of untreated wastewater from households, and industry and agricultural runoff have led to severe pollution of the aquatic environment[12,14,18,35–37]. In a few earlier water scarcity assessments for Chinese cities and river basins, the water quality issue was included by comparing the gray water footprint (i.e., the amount of water required to dilute pollutants in wastewater to meet environmental water quality standards) with water availability[22,24,34,38]. Later, China's water scarcity was analyzed at the national level based on this gray water footprint concept, however, using a rather coarse spatial resolution for 31 provincial-level administrative units[39]. A recent study assessed the implications of pollution for water scarcity by explicitly considering sectoral water quality requirements in comparison to available water quality[34].

Despite a recognized need, nationwide assessment of water quality as a contributing factor to water scarcity in China has not yet been implemented at a high spatial resolution, probably due to a limited coverage of water quality data[20,40]. The impact of inadequate water quality on water scarcity and its regional inequality remains unclear. While most large-scale water scarcity assessments were implemented at either grid cell or watershed scale[4,25–27,30–32,39], the effect of using different spatial resolutions on water scarcity and accompanied uncertainty still present a great knowledge gap.

To address these, we quantified China's present-day water scarcity, by examining needs for human water uses meeting both quantity and quality requirements at various temporal and spatial scales, in the meantime, taking into account environmental flow requirement (EFR). We compiled nationwide datasets consisting of water availability, water quality (measured by three typical water quality indicators, including the chemical oxygen demand, COD; ammonium nitrogen, $NH_4^+$-N; and electrical conductivity, EC), and sector-specific water withdrawal (for irrigation, industry, domestic use, and eco-environmental compensation use). All the datasets contain multiple geographic and temporal scales: at the $0.25 \times 0.25$ arc-degree grid cell, the first-order, second-order, and third-order basin levels on the annual, seasonal, and monthly basis for the 5-year period: 2012–2016 (Methods section). We then assessed the impact of inadequate water quality on water scarcity across four different geographic levels at three time scales. The results are crucially important for informing policy-making for regional water scarcity adaptation and alleviation.

## Results

**Quality-included water scarcity at various geographic scales.** Quantity-based water scarcity (referred to as WSqua, see Methods section), quality-based water scarcity (also pollution-induced water scarcity, abbreviated as WSpol), and water scarcity based on the combined effect of both quantity and quality (referred to as WScom) in present-day China exhibit geographical variations (Fig. 1). At the grid cell level, WScom tends to increase areas under water scarcity and intensify water scarcity in many places in comparison to WSqua (Fig. 1b–d). A total of 28.8% and 32.0% of China's area suffer WSqua and WScom (WSqua > 1 and WScom > 1, see methods for the equations), respectively. Water scarce areas are mainly distributed in North China. Over half areas in the Huai, Hai, Yellow, and Liao River basins, and 45.4% areas in the Songhua River basin are under WScom. In the Northwest River basin with large wild and unpopulated regions, water scarcity mainly occurs in west Xinjiang province with large irrigated croplands. While not under WSqua, a number of grid cells in the middle and lower reaches of Yangtze River and on the southeast coasts face WScom. This implies that in South China, water shortage is only relevant when quality issue is considered, though quality-induced water scarcity alone is not significant.

Water scarcity assessments at the basin levels, in which the heterogeneity of water availability, withdrawal, and quality within a basin is neglected, show some consistency with the high-resolution grid cell-based results. That is, basins with higher percentages of grid cells under water scarcity tend to be more water stressed (Fig. 2). The third-order, second-order, and first-order basins under WScom constitute respectively 64.6%, 59.2%, and 70.0% of the total numbers of basins of corresponding levels. At the first-order basin level, the four basins in South China are mostly under low WScom (WScom < 1, with the exception for the Southeast River basin slightly >1), whereas the other six basins in North China face both WSqua and WScom. In the most water-stressed Hai and Huai River basins with dense population and intensive agricultural activities, WSqua is >5, indicating that human water withdrawal (not accounting EFR) is larger than local available water resource. Areas where water withdrawal exceeds water availability have long been relying on exploitation of nonrenewable fossil groundwater and inter-basin water transfers, e.g., the south-to-north water transfer project[17,18]. Consequently, groundwater overexploitation has resulted in abrupt decline of groundwater tables, which is unsustainable and leads to a series of eco-environmental problems. Inter-basin water diversion is usually energy consuming and cost intensive, and may have adverse eco-environmental impacts on the source basins[17]. In the Hai River basin, which is home to nearly 10% of the Chinese population, the inclusion of water quality dimension has led to more than doubling the value of WScom in comparison to WSqua, indicating a significant effect of degraded water quality on exacerbating water scarcity.

**Seasonal and sectoral water scarcity.** As all the influencing factors for water scarcity, i.e., water resources availability, water withdrawal, and water quality show seasonal variations, water scarcity levels also present seasonal differences (Fig. 3). The climate in China is mostly controlled by the Pacific and Indian Ocean monsoons, so that in most regions 70–80% of annual precipitation falls between four consecutive months (i.e., May to August, or June to September[41]). Overall, spring is the most water scarce season, as the season is relatively dry (Supplementary

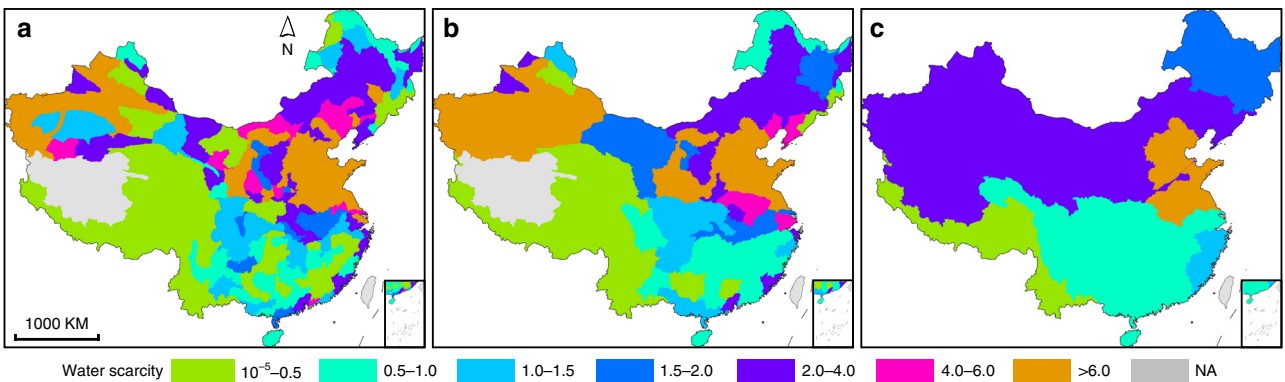

**Fig. 1 Estimates of annual water scarcity at the grid cell level in China. a** Spatial distributions of river basins in this study. **b** Quantity-based water scarcity (WSqua). **c** Pollution-induced water scarcity (WSpol). **d** Combined water scarcity, including both quality and quantity effects (WScom). Estimated water scarcity was based on the average of annual assessments during 2012–2016 at a spatial resolution of 0.25 × 0.25 arc-degree ($n = 15997$). The graph at the lower left corner in **a** represents the sampling locations of water quality. Maps of grid cell-level WScom at other time scales are shown in Supplementary Fig. 2. NA, no data or water scarcity is $<10^{-5}$. Source data are provided as a Source Data file.

**Fig. 2 Quality-included water scarcity (WScom) across river basins on an annual basis. a** The third-order basin level ($n = 209$). **b** The second-order basin level ($n = 76$). **c** The first-order basin level ($n = 10$). Maps of WScom at other time scales for these three basin levels are shown in Supplementary Figs. 3–5. Source data are provided as a Source Data file.

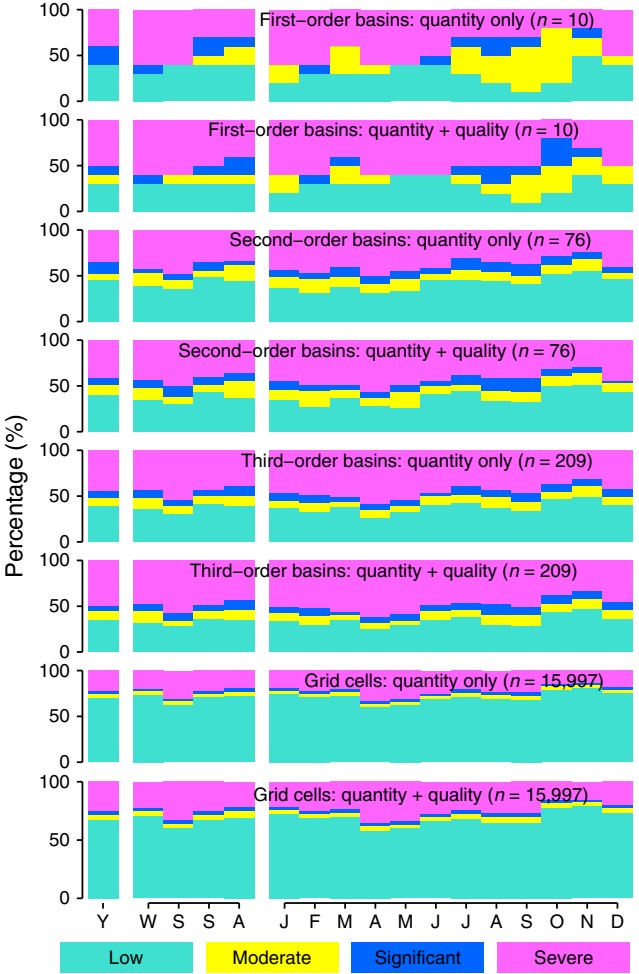

**Fig. 3 The proportions of regions under various water scarcity levels.**
Estimates were made across four different geographic scales at three temporal scales (Y, yearly; W–A, seasonal from winter throughout autumn; and J-D, monthly from January throughout December. Winter corresponds the months from December to February). The level of water scarcity is defined in Methods section. Source data are provided as a Source Data file.

Fig. 1), and the agricultural water demand is high in this early growing season. Seasonal water scarcity maps show significant spatial differences (Supplementary Figs. 2–5). WScom occurs in the middle reach of Yangtze River basin and lower reach of Southwest River basin in spring when a large quantity of irrigation water is needed, but not in other seasons. Many third-order basins in North China are under WScom in all the months throughout the year. A few third-order basins in South China, which do not face annual WScom, however, still suffer WScom in dry seasons or months. Figure 3 also shows that seasonal water scarcity is sensitive to the geographic scale. In the most water scarce spring, basins under WScom comprise 70.0%, 69.7%, and 71.3% of the total numbers of basins at the first-order, second-order, and third-order basin levels, respectively; 39.3% gird-cells suffer WScom. The season with the least number of geographic units under WScom is summer or autumn, depending on the geographic scale.

Proportions of sectoral water scarcity levels in the first-order basins are provided in Supplementary Fig. 6. Agriculture is the sector where water scarcity has the greatest relevance, because it is the most water consuming sector that represents ~67% of the total water withdrawal at the national level. On the annual basis, agricultural water withdrawal always comprises the largest share

of the total water withdrawal in the first-order basins. Agriculture-induced WScom is occasionally smaller than industry or domestic sector-induced WScom in 6.6% and 8.6% of the second-order and third-order basins, respectively. As a result of seasonal variations in sectoral water withdrawals (in particular in agricultural water withdrawal), sectoral water scarcity proportions also show great seasonality. Contributions of water scarcity from agricultural water withdrawal are usually most significant in spring and summer (Supplementary Fig. 6). Since domestic and industrial water uses usually have priority over agriculture when competition between sectoral water uses intensifies, agriculture is the most vulnerable sector to water scarcity[42–44]. However, due to its high water use volume and intensity, the potential for agricultural water conservation is also high. Therefore, one of the keys for addressing water scarcity is sustainable agricultural water use and management[9,45], while ensuring food security.

**Population under water scarcity at multiple scales.** In China, ~80% of the human population live in 10% of the land area. Water scarce regions are densely populated, because water demands are closely related to human activities, while some of these densely populated areas are also in drier parts of the country. Superimposing the population distribution map on the results of our water scarcity assessment enables the number of people living in water scarcity conditions to be estimated[19,26]. The numbers of people facing different water scarcity levels based on assessments at different temporal and geographic scales are shown in Fig. 4. On an annual basis, about 51.3%, 75.2%, 86.6%, and 86.1% of a total population of 1.36 billion in China live in WScom conditions according to the assessments based on the first-order, second-order, third-order basin, and grid cell scales, respectively (Fig. 4a). A higher spatial resolution estimation generally corresponds to a larger number of people facing water scarcity. On the first-order basin scale, people who face WScom live in the six basins in North China plus the Southwest River basin, while on the grid cell scale, people living in highly populated areas in other basins in South China also suffer WScom. The inclusion of WSpol in the assessment makes between 2.3–5.7% more people under WScom in China (in comparison to those facing WSqua), depending on different geographic scales.

When seasonal variability is considered, about 65.1%, 97.7%, 96.1%, and 92.9% of the population are under WScom for at least one season, with increasing geographic resolutions from the first-order basin to grid cell scales (Fig. 4b). A total of 100%, 99.9%, 98.8%, and 94.7% of the population live in WScom for at least 1 month, based on assessments of the above four geographic scales, respectively. These numbers are much larger than those from annual estimates, implying that annual estimates may underestimate water scarcity severity[26,29,31]. All of the ten first-order basins face WScom for at least 1 month. In the least water-stressed Southeast River basin, where the annual WScom is <0.1, WScom occurs in December when the monthly natural water resource availability is the least and pollution discharges in cities (e.g., Xining and Lhasa) are remarkable.

About 51.3%, 82.6%, 94.4%, and 91.6% of the population face severe WScom (WScom > 2) for at least 1 month according to assessments at the four geographic scales with increasing resolutions (Fig. 4c). The number of people under severe WSqua for at least 1 month (WSqua > 2) ranges between 0.62 and 1.24 billion. These estimates bracket a previous estimate of 0.9 billion from Mekonnen and Hoekstra's global study of water quantity[26], yet provide an uncertainty range originating from changing geographic scales. A total of 0.34–0.82 billion people are subject to severe monthly WScom all year round. Of all the people who face severe monthly WScom, 0.34 billion people live in five basins

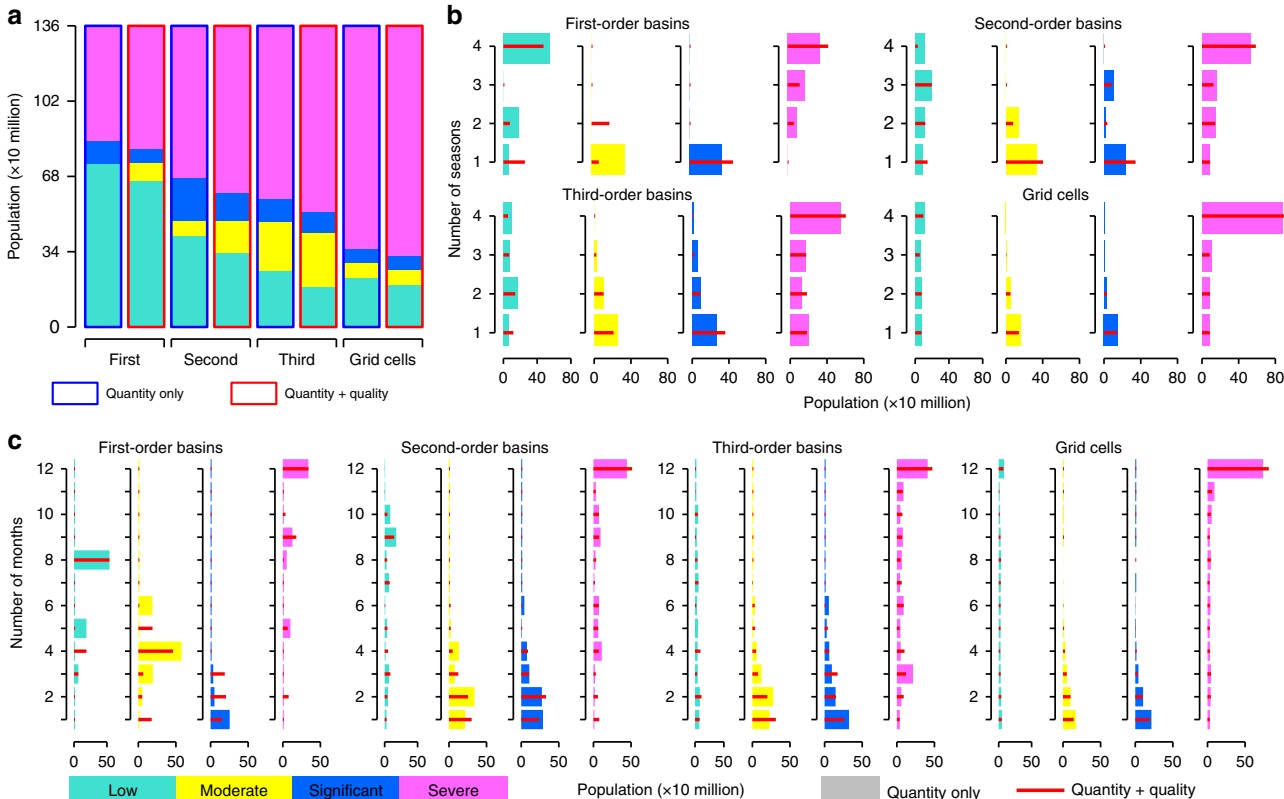

**Fig. 4 Estimated number of people living under different water scarcity levels.** Estimates were made according to quantity-based and quality-included water scarcity assessments across four geographic scales on three temporal scales. **a** Annual basis. **b** Seasonal basis. **c** Monthly basis. Source data are provided as a Source Data file.

in North China at the first-order basin level, while 0.82 billion people also include those who live in the Songhua River basin and South China according to the third-order basin scale assessment.

**Regional inequality of water scarcity**. As specified previously, water scarcity in basins in North China is generally more severe than South China. Given limited total available water resources (~2.77 trillion m³ annually on average in China[41]), a greater level of inequality corresponds to more severe water scarcity in a few regions and more people under water scarcity, if the water scarcity level is held constant at the national level. The curves of cumulative proportions of water withdrawal and water availability (sorted in an increasing order of the ratio of water withdrawal to water availability) on the annual basis are far from the 45 degree line, which represents perfect water withdrawal equality, confirming high inequality in spatial water scarcity in China (Fig. 5a). The curves taking into account water quality requirements are even further from the equality line than those considering only water quantity, implying that the inclusion of water quality dimension aggravates water scarcity inequality in China. On the graph for first-order basins, basins in South China are located on the bottom of curves where WSqua or WScom values are relatively low, whereas basins in North China are on the top of the curves with higher WSqua and WScom values. Graphs showing inequality of water scarcity for different time scales are shown in Supplementary Figs. 7–10.

The Theil's *L* index (Methods section) provides a quantitative measure for inequality of water scarcity levels at different temporal and geographic scales (Fig. 5b). A higher Theil's index corresponds to a greater inequality. The Theil's index for WScom is always higher than for WSqua. A finer resolution assessment for water scarcity corresponds to a higher Theil's index. The

Theil's indices also present seasonal variabilities. The greatest regional inequality occurs in the most water scarce spring. The month with the greatest regional water scarcity inequality falls between March and May, depending on the geographic scales. Because natural water bodies have a self-cleansing capacity that can remove polluting substances by a series of chemical and biological self-purification processes and such a capacity is often related to water availability, water pollution is also about quantities. Therefore, quality-induced regional inequality appears to be larger in high water scarce seasons or months. This also explains greater regional inequality in WScom than in WSqua.

## Discussion

This research provides a comprehensive analysis of China's present water scarcity levels at various geographic and temporal scales, for the first time including the implications of water quality. The results show that the inclusion of water quality in water scarcity assessment leads to aggravated water scarcity, as well as greater water scarcity inequality in China. North China often suffers from water scarcity from both insufficient water quantity and inadequate quality throughout the year, whereas South China is subject to seasonal water scarcity mainly due to water quality degradation. Chinese state and local government have invested thousands of billions of Chinese Yuan to enhance wastewater treatment and control pollution sources dedicated to environmental restoration in the recent decade[37]. Contributions of the investments on the improved surface water quality (characterized mainly by COD and $NH^+_4$-N concentrations) and the regional difference were quantified in a recent study[46]. Notwithstanding a recent trend in improved water quality, our results show that quality still presents a great issue for achieving safe water supply in China.

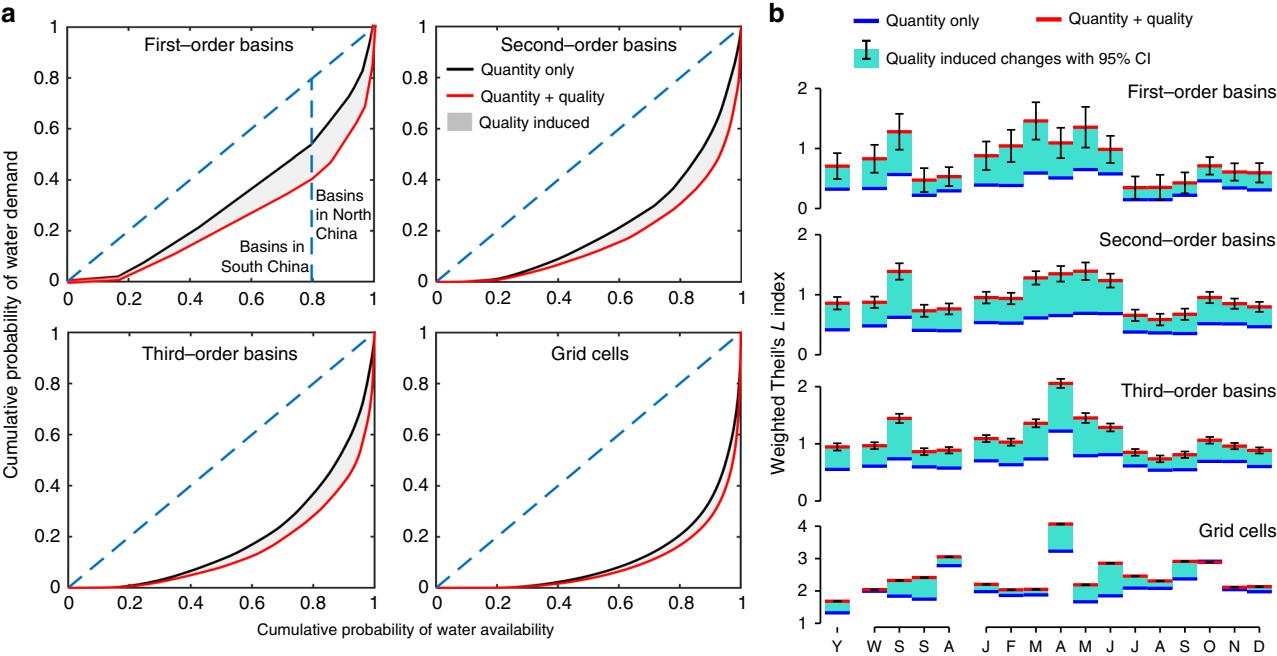

**Fig. 5 Regional inequality of water scarcity. a** Cumulative probability of water availability against cumulative portability of water withdrawals on the annual basis, sorted by increasing magnitudes of the ratio of water withdrawal to availability according to different geographic scales. **b** Disparities in water scarcity levels among regions. Theil's $L$ index is used to indicate regional water scarcity disparities, and the 95% confidence interval (CI) of Theil's $L$ index was calculated based on the normal approximation (Y, yearly; W–A, seasonal from winter throughout autumn; and J–D, monthly from January throughout December. Winter corresponds the months from December to February). Source data underlying **b** are provided as a Source Data file.

The metric WSpol used in this study for measuring quality-induced water scarcity, i.e., the ratio of water required for dilution to obtain adequate quality for sectoral water uses to water availability[34], is useful in guiding policy-making to identify basins where water supplies are mostly threatened by degraded water quality. At the third-order basin level (at which aquatic environment restorations are often planned and managed in China), prominent examples of water quality inferior to level IV (according to the Environmental Quality Standard for Surface Water in China[47]) include sub-basins in Hai River, Yellow River, and Liao River basins (Supplementary Fig. 11). Given that the investment budget into aquatic environment restoration is limited each year, our results suggest that the need for water quality improvement in the Hai River sub-basins is the most urgent due to the severe scarcity (WSpol > 2). The impact of water quality on domestic and production water uses can also be analyzed using the concept of water carrying capacity, which refers to the maximum population or economic scale that can be sustainably supported by available water resources[48].

The water carrying capacity, in terms of a multiplier of present-day population or economic scale, is analyzed in Supplementary Note 1, and similar conclusions can be derived as those from WSqua and WScom assessments. For instance, at the first-order basin level on an annual basis, the quality-included carrying capacities in all the six North China basins and the Southeast River basin is <1, indicating an insufficient capacity of water resources for sustaining the population and economic activities in these basins. While the quantity is sufficient for supporting human's water needs and EFR in the Southeast River basin, its inadequate water quality makes the quality-included carrying capacity <1 (Supplementary Note 1, Supplementary Fig. 12).

While most global water scarcity studies performed in single- or multi-model settings have estimated water uses based on FAO AQUASTAT country-specific data or model simulations[8,26,31,32], the regional-scale statistics used in this study allow for more accurate high-resolution sectoral water withdrawal downscaling in China. We estimate that the number of people facing WScom is between 0.70 and 1.36 billion, which presents large uncertainty when different temporal and geographic scales are considered. Most previous water scarcity assessments recommended a grid cell-based assessment, claiming that it is able to consider spatial variations of water resources and withdrawals within basins[3,26,32]. However, this high-resolution approach sometimes differs from the real world, because the distance between water abstraction and water use can be substantial and not captured within a grid cell[10]. For instance, large cities have extensive urban water supply systems, sometimes hundreds of kilometers distant from water sources[49]; many river basins (e.g., the Yellow River basin) have implemented integrated basin-scale water resources allocation in order to address sub-regional and sectoral water use competition. Previous studies also highlighted that annual assessments hide the intra-annual variability of both water availability and uses, and may underestimate the extent of water scarcity[19,26,29,32]. However, because reservoirs are common on Chinese rivers[8,10,50], occasionally seasonal or monthly water scarcity in many basins may have been addressed or reduced by seasonal flow regulations. Hence, assessment at a finer geographic or temporal resolution does not necessarily correspond to a better water scarcity estimate unless it takes account of infrastructure operation.

Therefore, we argue here that it is essential to conduct water scarcity analyses at different temporal and geographic scales, unless sufficient evidence supports a specific scale. Analyses at different geographic and temporal scales provide lower and upper bounds of water scarcity levels due to uncertainty in practical water uses and management. Our estimates of water scarcity intervals at various spatiotemporal scales also provide insights in the extent to which a corresponding measure (i.e., water resources allocations between seasons or sites) can be useful in coping with water scarcity. For instance, in basins where people suffer water scarcity in all the months, seasonal flow regulation will not be

effective in reducing water scarcity. Because human interventions (e.g., man-made reservoirs and human water uses), as well as legal and institutional arrangements may all result in changes in water scarcity[8], more detailed information from improved monitoring, and management reporting needs to be incorporated in future water scarcity assessment to reduce uncertainty.

The role of seasonal variability for EFR in water scarcity assessment has been highlighted in recent studies[39,51]. Here, we present seasonal and monthly water scarcity with a fixed pro-portion of EFR for consistency with annual analyses. The inclu-sion of seasonal variability for EFR has an effect on seasonal water scarcity (Supplementary Note 2, Supplementary Fig. 13a), because the proportion of EFR directly impacts water resources avail-ability for human uses. Nevertheless, regional inequality of monthly water scarcity is insensitive to seasonal variability for EFR in this case (Supplementary Fig. 13b). It has to be recognized that we do not differentiate groundwater withdrawal and quality from surface water in this research, because of a lack of nation-wide groundwater quality data. Quality from both surface water and groundwater for sectoral water uses is represented by in situ surface water quality. This will not have any impact on WSqua. However, in regions where surface water quality is inadequate for specific uses and a significant proportion of groundwater is exploited, WSpol is expected lower in reality, as groundwater quality is often superior to surface water. Our research, which does not focus on the effect of large engineering approaches on water scarcity, does not consider water scarcity condition changes due to water transfers across basins (e.g., south-to-north water transfer project).

Given the fact that over half of the population in China are currently facing WScom, even when seasonal and locally grid cell water scarcity (that can be solved by adequate seasonal within-basin water resources allocations) is not taken into account, water use efficiency improvement[41], water conservation[18,52], physical and virtual water transfers across basins[17,53], and aquatic environmental restoration[14,18,37] are probably the key solutions for alleviating water scarcity in China, which is essential for sustainable socio-economic development and eco-environmental protection. High regio-nal inequality of water scarcity in China urges locally specific policies for water demand management. In the meantime, inter-basin water transfers can directly mitigate water scarcity by supplying extra water sources to water scarce basins. Water scarcity assessment with a finer resolution corresponds to a greater inequality, indicating that water allocations within and between basins will be effective in reducing water scarcity inequality. However, the impact of water transfers on both source basins (e.g., social and eco-environmental impacts) and receiving basins (e.g., non-native species invasions and spreading of chemical/biological contaminants) needs to be carefully analyzed to minimize negative consequences.

The urgent need to further improve inland water quality in China, especially in northern basins, has been pointed out[14,35–37,54]—mostly because water pollution presents a great risk to public health and ecosystem services—and our results show evidence of this need from the perspective of providing adequate water supplies to population, economy, and ecosystems based on a quality-included water scarcity analysis.

The challenge of WScom that China faces is also shared by many other countries, as freshwater pollution is a worldwide problem in both developing and developed countries[39]. The methodology in this research could help to assess the mag-nitude of the challenge in other countries, thereby helping to formulate effective policies to achieve sustainable water supply. However, at present, inadequate water quality data availability makes it challenging to develop a global WScom assessment.

## Methods

**Assessment of water scarcity considering water quality.** For a target region, quantity-based water scarcity WSqua is measured as the ratio of regional water withdrawal to water availability, considering a balance between human uses of freshwater and ecosystem protection:

$$WSqua = \frac{\sum_i D_i}{Q - EFR} \quad (1)$$

where $D_i$ is water withdrawal for sector $i$, $Q$ is water availability, EFR is the environmental flow requirement, which is defined as 80% of water availability, following previous studies[19,26]. Adapting from van Vliet et al.[34], a dilution approach is applied to translate inadequate water quality into extra water quantity required so that it can be compared with water availability for water scarcity assessment. Quality-based water scarcity WSpol is calculated as the ratio of water required for dilution to obtain adequate quality for water uses to water availability:

$$WSpol = \frac{\sum_i dq_i}{Q - EFR} \quad (2)$$

with

$$dq_i = \max\left(dq_{i,j}\right)$$

$$dq_{i,j} = \begin{cases} 0, & C_j \leq Cmax_{i,j} \\ D_i\left(\frac{C_j}{Cmax_{i,j}} - 1\right), & C_j > Cmax_{i,j} \end{cases} \quad (3)$$

where $dq_i$ is extra water required for dilution to obtain acceptable quality for water use sector $i$, $dq_{i,j}$ indicates the amount of dilution water for sector $i$ based on water quality parameter $j$, $C_j$ is actual water quality level of parameter $j$, and $Cmax_{i,j}$ is the max-imum water quality threshold based on parameter $j$ for water use sector $i$. Equation (3) is slightly different from its original form developed by van Vliet et al.[34]. Instead of diluting all the available water to obtain water quantity needed for a sectoral water use, we estimate the extra amount of water by diluting the volume of the sectoral water withdrawal. Quality constraint for EFR is not considered in this study.

Combined water scarcity WScom taking into account both water quantity and quality dimensions is calculated:

$$WScom = WSqua + WSpol \quad (4)$$

In this study, water scarcity is classified into four levels based on the value of WSqua and WScom: low (<1.0), moderate (1.0–1.5), significant (1.5–2.0), and severe (>2.0), which is consistent with many previous studies[19,26,39,55]. The mean of WSqua or WScom values in 5 years from 2012 to 2016 is reported to represent the present-day water scarcity levels in China, because high-density water quality monitoring data are only available in this recent period. The inter-annual variabilities of WSqua and WScom are thus not discussed in this study.

**Water quality requirements for sectoral water uses.** Water withdrawal is mainly for four major sectoral uses in China: agriculture, industry, domestic uses, and eco-environmental compensation. Quality requirements for each sectoral water use Cmax in Eq. (3) are defined according to three typical water quality measures—COD (the permanganate index), $NH_4^+$-N, and EC. For agricultural water use, EC = 0.7 dS m$^{-1}$, which indicates a salinity constraint for crops, sug-gested by FAO[56] is considered as the maximum water quality threshold for irri-gation. Maximum water quality thresholds for quality requirements for other sectoral uses are based on COD and $NH_4^+$-N concentrations according to the Environmental Quality Standard for Surface Water in China[47]: COD = 6.0 and $NH_4^+$-N = 1.0 mg L$^{-1}$ for domestic uses (water quality level I–III), COD = 10.0 and $NH_4^+$-N = 1.5 mg L$^{-1}$ for industrial uses (water quality not inferior to level IV), and COD = 15.0 and $NH_4^+$-N = 2.0 mg L$^{-1}$ for eco-environmental compensation (water quality not inferior to level V). These thresholds of water quality parameters represent the minimum water quality requirements for sector-specific uses.

**Data sources and data processing.** Annual provincial-level natural water avail-ability data from 2012 to 2016 (ref. [57]) are downscaled to monthly grid cell values based on Variable Infiltration Capacity (VIC) hydrologic model simulation results[58]. The VIC simulation results, which have been validated with gauge measurements across China, provide a grid cell-level estimate for monthly water availability (including both subsurface and surface runoff) with a spatial resolution of 0.25 × 0.25 arc-degree. The annual statistical data of water availability are firstly downscaled to the grid cell level on the annual basis by letting water resources in grid cells pro-portional to those from VIC model simulation results (Supplementary Fig. 14a). The grid cell-level annual water availability is then disaggregated into monthly values proportional to simulated monthly runoff in corresponding grid cells. The average annual freshwater availability in the period 2012–2016 in mainland China is ~2.91 trillion m$^3$, ~5% more than the long-term average ~2.77 trillion m$^3$ and can roughly represent the average annual water resource availability condition in China.

Annual sectoral water withdrawal data at the province-level in China for the period 2012–2016 (ref. [57]) are downscaled to monthly grid cell data based on multisourced information. Agricultural water withdrawal is mainly for irrigation, and is hence disaggregated based on relevant information including crop land uses

(Supplementary Fig. 14b) and net irrigation requirements. Net irrigation requirements are calculated as the difference between the reference crop evapotranspiration and effective precipitation at the grid cell level, which are derived mainly from meteorological data. Annual industrial water withdrawal is downscaled based on maps of the industrial gross domestic product (GDP) at the grid cell level (Supplementary Fig. 14c). The maps of industrial GDP are generated by disaggregating province-level data through a proportional sharing method based on areas of industrial lands weighted by nighttime brightness[59]. Industrial water withdrawals are assumed uniformly distributed in all the months within a year. Annual domestic water withdrawal data are disaggregated according to spatial urban and rural population distributions and monthly water use factors. Rural population is disaggregated based on rural residential areas in grid cells, and urban population is disaggregated based on urban areas in grid cells weighted by nighttime light (similar to industrial GDP downscaling) (Supplementary Fig. 14d). A monthly factor for domestic water use, which is a function of temperature, is introduced to consider seasonal water use variations following previous studies[32,60]. Eco-environmental compensation water withdrawal is mainly used for irrigating green spaces, and replenishing dry rivers and lakes in urbanized areas[57], and only represents 2% of the total water withdrawal in China. As it is difficult to discern green spaces and water bodies in urbanized areas based on the available land use maps, we assume that water withdrawal for eco-environmental compensation is proportional to the size of urbanized areas in grid cells (Supplementary Fig. 14e). Eco-environmental compensation water withdrawals are assumed uniformly distributed in all the months within a year (see Supplementary Notes 3 and 4 for detailed methodology and equations for sectoral water withdrawal downscaling).

Water quality data are collected from the national environmental monitoring network, which covers China's major inland rivers and lakes. Monthly observations of COD, $NH^+_4$-N, and EC are available for 2630 sampling sites (with an average spatial density of 2.74 sampling sites per $10^4\,km^2$, Fig. 1a) for the period 2012–2016 (Supplementary Note 5, Supplementary Table 1). Based on the regional assemblage of site-level measurements, we use an inverse distance weighting function with an exponent of 1 to interpolate the spatial water quality parameters[61] at the grid cell level. We assemble site-level water quality data in second-order basins in which the majority (~75%) of grid cells are covered, and make the spatial interpolation for these grid cells. For the remaining ~25% of grid cells, the spatial interpolation of water quality parameters is made within first-order basins due to the lack of site-level observations in corresponding second-order basins. The cross-validation, which estimates water quality in grid cells containing sampling sites based on measurements in other grid cells, is applied to test the validity of the interpolated water quality parameters by comparing with the actual measurements. The results show that the interpolation provides sufficiently accurate water quality estimations with high R-square and low root-mean-square error (RMSE; $R^2 = 0.81$ and RMSE = 1.11 mg L$^{-1}$ for COD; $R^2 = 0.80$ and RMSE = 1.13 mg L$^{-1}$ for $NH^+_4$-N; and $R^2 = 0.90$ and RMSE = 0.74 dS m$^{-1}$ for EC). At the grid cell level, the mean value of a monthly water quality parameter in one season or 1 year (weighted by monthly water availability) is used to represent the corresponding seasonal or annual water quality. Water quality at a basin level is represented by the mean of water quality parameters in grid cells within corresponding basins (weighted by grid cell water availability).

**Weighted Theil's L Index for regional inequality**. We used the Theil's L index $T_L$, i.e., the mean log deviation, to measure the inequality of water scarcity among $n$ regions. $T_L$ is weighted by water withdrawal:

$$T_L = \sum_{i=1}^{n} \left[ w_i \times \mathrm{Log}\left( \frac{\mu}{\mathrm{WS}_i} \right) \right] \quad (5)$$

with

$$\mu = \sum_{i=1}^{n} (w_i \times \mathrm{WS}_i)$$

$$w_i = \frac{\mathrm{WW}_i}{\sum_{i=1}^{n} \mathrm{WW}_i}$$

where $\mathrm{WS}_i$ is the water scarcity index (either WSqua or WScom) in geographic unit $i$, and $\mathrm{WW}_i$ is the amount of water withdrawal in geographic unit $i$.

**Reporting summary**. Further information on research design is available in the Nature Research Reporting Summary linked to this article.

## Data availability
Detailed information regarding the data that support the findings of this study can be found in Supplementary Table 1. The source data underlying Figs. 1b–d, 2–4, and 5b, and Supplementary Figs. 1–6, 12, and 13 are provided as a Source Data file. The additional data that support the findings of this study are available from the corresponding author upon reasonable request.

## Code availability
All computer codes used in the current study are available upon reasonable request.

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

## Acknowledgements

This work was jointly supported by the National Key Research and Development Program of China (2017YFB0503600), the Strategic Priority Research Program of the Chinese Academy of Sciences (XDA20040401), the National Natural Science Foundation of China (41421001, 41525004, and 41771418), and the Key Research Program of Frontier Science of the Chinese Academy of Sciences (QYZDY-SSW-DQC007).

## Author contributions

T.M., S.S., C.F., and C.Z. designed research, T.M., S.S., Y.N., L.H., J.Y., N.Z., Y.D., T.P., W.C., and C.S. performed research, T.M., S.S., Y.N., and C.Z. analyzed data, T.M., S.S., G.F., and J.W.H wrote the manuscript, and all coauthors contributed to the interpretation of the results and to the text. All authors read the manuscript and approved the submission.

## Competing interests

The authors declare no competing interests.

## Additional information

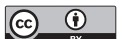

