## [Peer Review File · Nature Communications]

Reviewers' comments:

Reviewer #1 (Remarks to the Author):

Pollution exacerbates China's water scarcity and its regional inequality

This study shows estimates of water scarcity including both water quantity and water quality for China and highlights the relevance of using different spatial scales (basin, subbasin order, grid) and temporal scales (annual, seasonal, monthly). The authors address an important topic that is interesting for a broad scientific audience. I think the results are interesting and the paper is overall well-written, but I have some comments that are listed below:

- Novelty: "Here we report a first nationwide water scarcity assessment in China, which addresses quantity and quality requirements for human water use and ecosystems at various temporal and geographic scales" (line 26-28). >> I think it actually not the first study presenting results of water scarcity including water quality for China (please see figure 2 of publication of Liu et al. (2017)). Also the indicator presented of water scarcity is strongly based on earlier work as mentioned, but I think the results of presenting water scarcity at different spatial and temporal scales are very interesting and an addition to the previous studies. This should be made more clear in the framing of this paper and abstract.
- More details should be given to ensure that the results can be reproduced by other researchers and to show the robustness of the results. More information in particular in the supplementary information is helpful regarding the following points, e.g. datasets used, the quality of these input data used for the water scarcity assessment, section of water quality parameters, sampling frequency and station density of the water quality observations, conversion of station water quality to gridded calculations of water scarcity, etc. Also more details are needed in terms of the quality of these datasets.
- It is unclear from how water quality for ecosystem health estimated. Different terminology is also used 'environmental flows for ecosystems, meeting both quantity and quality' 'environmental flow requirement' and 'eco-system compensation'. I think the environmental flow from water quantity perspective (80% of water availability) is a bit too coarse approach in particular because this study accounts for different temporal scales, I would recommend an approach that accounts for seasonal flow variability.
- Some assumptions have been made but with limited explanation, e.g. definition of water scarcity classes. In this study water scarcity (quantity) levels of 0.4 or higher are indicated by low water scarcity while other studies defined water stress of 0.4 as severe water scarcity (Hanasaki et al., 2018; Liu et al., 2017). This discrepancy requires some clarification.
- Three water quality parameters are included (chemical oxygen demand, ammonium nitrogen and electrical conductivity) to represent water quality status but why was this selection made? This is not explained in the main text, methods or supplementary, but it can likely has strong impacts on your results.
- Was not fully clear to me from the description to what extent groundwater extractions and groundwater availability and quality are incorporated.
- Some figures are pretty hard to read in particular figure 2 and 3. I would suggest to change color bar (e.g. to green yellow red) to improve the readability and interpretation of the results (as blue and green look almost similar in my printed version) and limit the use of abbreviations (e.g. FOB, SOB etc).

Additional references

Hanasaki, N., Yoshikawa, S., Pokhrel, Y., Kanae, S. (2018) A Quantitative Investigation of the Thresholds for Two Conventional Water Scarcity Indicators Using a State-of-the-Art Global Hydrological Model With Human Activities. *Water Resources Research* 54, 8279-8294.

Liu, J., Yang, H., Gosling, S.N., Kummu, M., Flörke, M., Pfister, S., Hanasaki, N., Wada, Y., Zhang,

X., Zheng, C., Alcamo, J., Oki, T. (2017) Water scarcity assessments in the past, present and future. *Earth's Future* 5, 545–559.

Reviewer #2 (Remarks to the Author):

Review of NCOMMS-19-29361

Reviewer recommendation: Revise and resubmit

The authors have tackled an important question: how does water quality, negatively impacted by pollution, impact water availability and scarcity at different geographical scales? The question is asked in the context of China, which faces acute shortages of water (quantity-based scarcity) as well as serious problems with water pollution. The author's assemble a large amount of data to conduct the analysis, and the findings are interesting. For this reason, I believe that the manuscript contains merit and in revised form might make a contribution to *Nature Communications*. However, there are several points that the authors should address in a revised manuscript:

- 1) What is the role of water-treatment in affecting the impact of water quality? Different cities may decide to spend more or less money and resources to remediate pollution. How do investments in improved water quality fit in?
- 2) The authors mention that groundwater cannot be included because of lack of data. This is a serious omission. Are there reasons not to consider using GRACE data for the more aggregated geographical scales? If the author's have a good reason not to consider the groundwater data, I believe that they must still discuss the role of groundwater more than they do currently.
- 3) The author's shorthand WSq, WSp, and WSc, are not intuitive and are difficult to keep track of.
- 4) Fig 3 is truly amazing and informative. It would be good for the authors to discuss this in more depth
- 5) The methodological approach using dilution is not explained to the reader, and should be.
- 6) The policy implications are not clear. For example, would the authors recommend more investment in interregional water transfers? E.g. China's South-North water transfer project? How does the discussion of variation across multiple geographic and temporal scales influence the larger takeaways? Influence policy recommendations?

Minor points:

- The writing is not as clear as it could be and should be copyedited for grammar and style
- The headline range of the number to be affected (.7-1.36 billion) is too large to be informative -- more attention to multiple-scale approach would be better
- The argument for analysis at multiple temporal and geographic scales is important, the authors could bring this to the forefront

Reviewer #3 (Remarks to the Author):

This study is useful since authors have highlighted the importance of quality of water although

quality and quantity are equally important. The work involves considerable data compilation from different sources as enumerated. The spatial presentation of results nationwide is highly useful for policymakers. The use of simple statistics is highly appreciated. The broad generalization arrived at spatially sounds a bit simplistic for many reasons. I would have liked to see an atlas of the agricultural carrying capacity of water on the basis of water quality; industrial carrying capacity on the basis of water quality and also human and animal habitat carrying capacity on the basis of quality of water. Then how quality coupled with the quantity of water can be used to modify the carrying capacity. Another aspect is water quality study need to be connected with the water usage pattern which presents different scenarios. Water for cooking and drinking have different quality criteria than washing and flushing the toilet. You can also look at such usage needs in industry and agriculture. It will be useful if you highlight with the relevant database the issue of carrying capacity and usage pattern

Responses to reviewers

NCOMMS-19-29361

Title: Pollution exacerbates China's water scarcity and its regional inequality

We sincerely thank all the reviewers for their critical reading and constructive comments and suggestions for improving our manuscript. We have carefully addressed all the comments and questions. Answers are typed in blue. Following the reviewers' comments and suggestions, we have revised the manuscript accordingly, as detailed below.

Reviewer #1 (Remarks to the Author)

This study shows estimates of water scarcity including both water quantity and water quality for China and highlights the relevance of using different spatial scales (basin, subbasin order, grid) and temporal scales (annual, seasonal, monthly). The authors address an important topic that is interesting for a broad scientific audience. I think the results are interesting and the paper is overall well-written, but I have some comments that are listed below:

Author responses:

We would like to sincerely thank the reviewer for providing encouraging and constructive feedbacks to our work. Please find detailed point-by-point responses below.

- Novelty: "Here we report a first nationwide water scarcity assessment in China, which addresses quantity and quality requirements for human water use and ecosystems at various temporal and geographic scales" (line 26-28). >> I think it actually not the first study presenting results of water scarcity including water quality for China (please see figure 2 of publication of Liu et al. (2017). Also the indicator presented of water scarcity is strongly based on earlier work as mentioned, but I think the results of presenting water scarcity at different spatial and temporal scales are very interesting and an addition to the previous studies. This should be made more clear in the framing of this paper and abstract.

Author responses:

We agree with the reviewer that the expressions mentioned above are inaccurate, and we have rephrased relevant sentences in the revised manuscript. In the abstract, we have revised the sentences: "Here we report a comprehensive nationwide water scarcity assessment in China, which explicitly includes quality requirements for human water uses." (Page 2, lines 25-26) As suggested by the reviewer, we also highlighted our analysis at multiple temporal and geographic scales in the abstract: "We highlight the necessity of implementing water scarcity assessment at multiple temporal and geographic scales." (Page 2, lines 26-27)

In the framing of this paper in the Introduction section, we have rephrased the sentences:

"In a few earlier water scarcity assessments for Chinese cities and river basins, the water quality issue was included by comparing the grey water footprint (i.e. the amount of water required to dilute pollutants in wastewater to meet environmental water quality standards) with water availability^{22,24,34,38}. Later, China's water scarcity was analyzed at the national level based on this grey water footprint concept, however, using a rather coarse spatial resolution for 31 provincial-level administrative units³⁹," (Page 2, lines 52-57)

“Despite a recognized need, nationwide assessment of water quality as a contributing factor to water scarcity in China has not yet been implemented at a high spatial resolution, probably due to a limited coverage of water quality data^{20,40}. The impact of inadequate water quality on water scarcity and its regional inequality remains unclear. While most large-scale water scarcity assessments were implemented at either grid cell or watershed scale^{4,25-27,30-32,39}, the effect of using different spatial resolutions on water scarcity and accompanied uncertainty still present a great knowledge gap.” (Pages 2-3, lines 59-64)

- More details should be given to ensure that the results can be reproduced by other researchers and to show the robustness of the results. More information in particular in the supplementary information is helpful regarding the following points, e.g. datasets used, the quality of these input data used for the water scarcity assessment, section of water quality parameters, sampling frequency and station density of the water quality observations, conversion of station water quality to gridded calculations of water scarcity, etc. Also more details are needed in terms of the quality of these datasets.

Author responses:

A table listing all the data sources used for the water scarcity assessment has been added in the Supplementary Information, with data description, resolutions, data sources and remarks on data quality are included (See Table S1 in Supplementary Information). Sampling frequency and monitoring station density of the water quality observations are provided in Table S1 and the main text now:

“Water quality data were collected on a weekly basis, and averages of the weekly measurements were used to represent monthly water quality conditions. The three water quality indicators were measured according to “Technical Specifications Requirements for Monitoring of Surface Water and Wastewater in China”, issued by Ministry of Ecology and Environment, for guidance.” (Table S1) “Monthly observations of COD, NH₄⁺-N and EC are available for 2630 sampling sites (with an average spatial density of 2.74 sampling sites/10⁴km², see Fig. 1a) for the period 2012-2016.” (Page 10, lines 367-370)

The process of converting site-level water quality measurements to gridded data is given in detail in the main text:

“Based on the regional assemblage of site-level measurements, we use an inverse-distance weighting function with an exponent of 1 to interpolate the spatial water quality parameters⁵⁶ at the grid cell level. We assemble site-level water quality data in second-order basins in which the majority (~75%) of grid cells are covered, and make the spatial interpolation for these grid cells. For the remaining ~25% of grid cells, the spatial interpolation of water quality parameters is made within first-order basins due to the lack of site-level observations in corresponding second-order basins.” (Page 10, lines 370-376)

In addition, the conversion results were validated by the cross-validation method, which are provided in the main text:

“The cross-validation, which estimates water quality in grid cells containing sampling sites based on measurements in other grid cells, is applied to test the validity of the interpolated water quality parameters by comparing with the actual measurements. The results show that the interpolation provides sufficiently accurate water quality estimations with high R-square and low root-mean-square error (RMSE) ($R^2 = 0.81$ and RMSE = 1.11 mg L⁻¹ for COD; $R^2 = 0.80$ and

RMSE = 1.13 mg L⁻¹ for NH₄⁺-N; R² = 0.90 and RMSE = 0.74 dS m⁻¹ for EC).” (Page 10, lines 376-381)

- It is unclear from how water quality for ecosystem health estimated. Different terminology is also used ‘environmental flows for ecosystems, meeting both quantity and quality’ ‘environmental flow requirement’ and ‘eco-system compensation’. I think the environmental flow from water quantity perspective (80% of water availability) is a bit too coarse approach in particular because this study accounts for different temporal scales, I would recommend an approach that accounts for seasonal flow variability.

Author responses:

In this study, water quality constraint is explicitly considered for human water uses, but not for environmental flow requirement. The main reason is two-fold: (1) water quality constraint for environmental flow requirement may result in negative water scarcity values (see Eq. c2 below), which do not have any physical meaning; (2) We believe that in most places the water quality constraint for human sectoral water uses (e.g. domestic and industrial uses) is much more stringent than that for environmental flow requirement. Quality based water scarcity can thus be well revealed by the indicator that considers quality constraint for sectoral human water uses. In this manuscript, the combined water scarcity WScom, taking into account both water quantity and quality dimensions for human water uses, is calculated as:

$$WScom = WSqua + WSpol = \frac{\sum_i D_i + \sum_i dq_i}{Q - EFR} \quad (c1)$$

where WSqua and WSpol represent quantity and quality (pollution) based water scarcity, respectively, D_i is water withdrawal for sector i , dq_i is extra water required for dilution to obtain acceptable quality for water use sector i (see Eq. 3 in the main text for the calculation formula), Q is water availability, EFR is the quantity for environmental flow requirement. If quality constraint for environmental flow requirement is considered, Eq. (c1) is turned to:

$$WScom = WSqua + WSpol = \frac{\sum_i D_i + \sum_i dq_i}{Q - EFR - dq_{EFR}} \quad (c2)$$

where dq_{EFR} is extra water quantity required for dilution to obtain acceptable quality for environmental flow requirement. The denominator in Eq. (c2) is possibly a negative value if water quality is inadequate.

We have revised the corresponding sentence in the introduction to state this:

“To address these, we quantified China’s present-day water scarcity, by examining needs for human water uses meeting both quantity and quality requirements at various temporal and spatial scales, in the meantime, taking into account environmental flow requirement (EFR).” (Page 3, lines 65-67)

We agree with the reviewer that different terminologies lead to confusion. We have used the same terminology “environmental flow requirement” for indicating the quantity of water required to maintain eco-environmental health. However, in China, a part of human water withdrawal is used for irrigating green spaces and replenishing dry rivers and lakes in urbanized areas, which is one of the four sectoral water uses in China’s water use statistics, and is referred as

eco-environmental compensation water use (Page 10, lines 359-361). Quality constraint for this eco-environmental compensation water use is considered.

We agree with the reviewer that environmental flow requirement from water quantity perspective (80% of water availability) is a bit too coarse approach, in particularly for seasonal water scarcity evaluations. Defining environmental flow requirement at the national scale in China is challenging, as there has been limited research discussing eco-hydrological responses to flow alteration at different temporal and geographic scales in China. For water scarcity assessment on an annual basis, 80% of water availability is considered for environmental flow requirement, following many previous studies (e.g. Hoekstra et al. 2012; Mekonnen and Hoekstra, 2016). In addition, the classification between low water scarcity and water scarcity using the criterion $WS_{qua} = 1$ corresponds to the traditional presumptive standard that depletion beyond 20% of natural flow increases risks to ecological health and ecosystem services (e.g. Wada et al. 2011). For consistence, seasonal water scarcity is also estimated assuming 80% of water availability for environmental flow requirement in the main text of this study.

We agree with the reviewer that the seasonal variability needs to be considered for environmental flow requirement. We have added an analysis that considers seasonal variability for environmental flow requirement referring to Pastor et al. (2014) (see Supplementary Information). New sentences have been added to discuss the effect of seasonal variability for environmental flow requirement on water scarcity assessment results in the Discussion section:

“The role of seasonal variability for EFR in water scarcity assessment has been highlighted in recent studies^{14,15}. Here we presented seasonal and monthly water scarcity levels with a fixed proportion of EFR for consistency with annual analyses. The inclusion of seasonal variability for an EFR has an impact on the seasonal water scarcity (Supplementary Fig. S13a) because the proportion of EFR directly impacts the availability of water resources for human uses.” (Page 7, lines 256-261). The approach for considering variable EFR and results are given in detail in Supplementary Information (Pages 6-7, lines 102-126 in Supplementary Information).

- Some assumptions have been made but with limited explanation, e.g definition of water scarcity classes. In this study water scarcity (quantity) levels of 0.4 or higher are indicated by low water scarcity while other studies defined water stress of 0.4 as severe water scarcity (Hanasaki et al., 2018; Liu et al., 2017). This discrepancy requires some clarification.

Author responses:

In this study, environmental flow requirement (EFR) is considered as 80% of water availability, and is deducted from available water resource in the denominator to calculate water scarcity indicators. Water quantity based water scarcity is defined (see Eq. 1 in the main body of the article):

$$WS_{qua} = \frac{\sum_i D_i}{Q - EFR} \quad (c3)$$

where D_i is water withdrawal for sector i , Q is total water availability. Water scarcity with WS_{qua} of 2 or higher is indicated by severe water scarcity in this study. Previous studies defined water scarcity indicator WS' as the ratio of water use to total water availability without explicitly considering EFR:

$$WS' = \frac{\sum_i D_i}{Q} \quad (c4)$$

Severe water stress is indicated by $WS' \geq 0.4$ or higher, as mentioned by the reviewer (Hanasaki et al., 2018; Liu et al., 2017). These two definitions of severe water scarcity, i.e. WSqua by Eq. c3 above 2 and WS' by Eq. c4 above 0.4, are actually consistent:

$$WS_{qua} = \frac{\sum_i D_i}{Q - 0.8Q} = \frac{\sum_i D_i}{0.2Q} = 5WS' \quad (c5)$$

This has been stated in the revised manuscript:

“In this study, water scarcity is classified into four levels based on the value of WSqua and WScom: low (< 1.0), moderate (1.0 - 1.5), significant (1.5 - 2.0) and severe (> 2.0), which is consistent with many previous studies^{19,26,39,55}.” (Page 9, lines 314-316)

- Three water quality parameters are included (chemical oxygen demand, ammonium nitrogen and electrical conductivity) to represent water quality status but why was this selection made? This is not explained in the main text, methods or supplementary, but it can likely have strong impacts on your results.

Author responses:

Quality influences water suitability for a specific use. There have been a number of different water quality guidelines related to different water uses, which do not show complete consistency because of the wide variability in field conditions. Our water scarcity assessment at such a large national scale requires effective but practical water suitability guidelines, in particular considering limited availability of water quality measurements at the national level. In China, freshwater usability for industrial, domestic and ecological compensation uses is usually roughly determined based on the classification of water quality (i.e. Classes I, II, III, IV, V and inferior V), according to Environmental quality standards for surface water (Ministry of Ecology and Environmental Protection of China, 2002). Though the classification of water quality levels should be made based on a wide range of parameters if available, in practice chemical oxygen demand and ammonium nitrogen are widely available and are commonly used for classification of water quality levels in China. Other parameters, such as concentrations of phosphorus, heavy metal and pesticide, are not widely available at the national level. The suitability of water for agriculture is mostly linked to the type and quantity of dissolved salts, and is thus determined by electrical conductivity in this study, with reference to FAO’s standards (Ayers & Westcot, 1985). Albeit not comprehensive, the selected three parameters are the main water quality parameters that are widely monitored at the national level and can generally well represent the characteristics of a water supply for its suitability for specific uses. Therefore, the selection of water quality parameters is based on data availability and best practices in China.

This has been stated clearly in a new section entitled “Water quality guidelines for sectoral water uses” in Supplementary Information in the revised version:

“Quality influences the water suitability for a specific use. There have been a number of different water quality guidelines related to different water uses, which are not wholly consistent because of the wide variability in field conditions. Our water scarcity assessment, done at a national scale, requires effective but practical water suitability guidelines, particularly considering

the limited availability of water quality measurements at the national level. The selection of water quality parameters indicating water suitability for specific uses is based on both data availability and relevance. In this study, the quality requirements for sectoral water uses were based on three standard water quality measures: chemical oxygen demand (COD), ammonium nitrogen (NH₄⁺-N), and electrical conductivity (EC). Although these quality requirements are not comprehensive, these three parameters are the main water quality indicators that are widely available at the national level, and generally, can well represent the characteristics of a water supply for its suitability for specific uses. The maximum water quality thresholds for each specific sectoral use, which are based on environmental quality standards for surface water in China and on FAO guidelines, are provided in the main body of this article.” (Page 4, lines 12-39 in Supplementary Information)

-Was not fully clear to me from the description to what extent groundwater extractions and groundwater availability and quality are incorporated.

Author responses:

Quantity based water scarcity WSqua is assessed by the ratio of the total water use to water availability deducted with EFR by the above Eq. c3. As the water availability including both surface and ground water resources, the separation of surface and ground water uses will not change the result for WSqua assessment. The quality based (pollution-induced) water scarcity WSpol is assessed by:

$$WSpol = \frac{\sum_i dq_i}{Q - EFR} \quad (c6)$$

where dq_i is extra water required for dilution to obtain acceptable quality for water use sector i , which is a function of water quality of a water source and quality requirement for a specific water use sector i :

$$dq_i = \max(dq_{i,j})$$

$$dq_{i,j} = \begin{cases} 0, & C_j \leq Cmax_{i,j} \\ D_i \left(\frac{C_j}{Cmax_{i,j}} - 1 \right), & C_j > Cmax_{i,j} \end{cases} \quad (c7)$$

where $dq_{i,j}$ indicates the amount of dilution water for sector i based on water quality parameter j , C_j is actual water quality level of parameter j , and $Cmax_{i,j}$ is the maximum water quality threshold based on parameter j for water use sector i , respectively. In theory, if a sectoral water supply is from groundwater, C_j in Eq. c7 should be represented by groundwater quality indicator. However, due to a lack of nationwide groundwater quality data, we do not differentiate surface and groundwater withdrawals, and the water quality is always indicated by surface water quality measurements. This will not have any impact on WSqua. However, since groundwater quality is often better than surface water, WSpol is expected to be lower in reality where surface water quality is inadequate for sectoral water uses and a significant proportion of groundwater is exploited. Our estimates therefore provide an upper bound for WSpol.

We have recognized this limitation in the Discussion section in the main body of the revised manuscript:

“Quality from both surface water and groundwater for sectoral water uses is represented by in-situ surface water quality. This will not have any impact on WSqua. However, in regions where surface water quality is inadequate for specific uses and a significant proportion of groundwater is exploited, WSpol is expected lower in reality, as groundwater quality is often superior to surface water.” (Page 7, lines 263-266)

- Some figures are pretty hard to read in particular figure 2 and 3. I would suggest to change color bar (e.g. to green yellow red) to improve the readability and interpretation of the results (as blue and green look almost similar in my printed version) and limit the use of abbreviations (e.g. FOB, SOB etc).

Author responses:

As the reviewer suggested, we have changed the color bar in Figures 3 and 4 in the revised manuscript. We guess that the reviewer meant Figures 3 and 4, rather than Figure 2 and 3. We have written the full terms instead of the abbreviations in the figures in the revised manuscript.

Additional references

Hanasaki, N., Yoshikawa, S., Pokhrel, Y., Kanae, S. (2018) A Quantitative Investigation of the Thresholds for Two Conventional Water Scarcity Indicators Using a State-of-the-Art Global Hydrological Model With Human Activities. *Water Resources Research* 54, 8279-8294.

Liu, J., Yang, H., Gosling, S.N., Kummu, M., Flörke, M., Pfister, S., Hanasaki, N., Wada, Y., Zhang, X., Zheng, C., Alcamo, J., Oki, T. (2017) Water scarcity assessments in the past, present and future. *Earth's Future* 5, 545–559.

Author responses:

Suggested references have been cited in the revised manuscript.

Reviewer #2 (Remarks to the Author)

The authors have tackled an important question: how does water quality, negatively impacted by pollution, impact water availability and scarcity at different geographical scales? The question is asked in the context of China, which faces acute shortages of water (quantity-based scarcity) as well as serious problems with water pollution. The author's assemble a large amount of data to conduct the analysis, and the findings are interesting. For this reason, I believe that the manuscript contains merit and in revised form might make a contribution to Nature Communications. However, there are several points that the authors should address in a revised manuscript:

Author responses:

We would like to sincerely thank the reviewer for the overall encouraging feedback to our study. All the constructive comments and suggestions have been taken into account and addressed. Please find detailed point-to-point responses below.

1) What is the role of water-treatment in affecting the impact of water quality? Different cities may decide to spend more or less money and resources to remediate pollution. How do investments in improved water quality fit in?

Author responses:

Because one of the main water pollution sources in China is domestic wastewater discharges and combined sewer overflows, as reported in recent water quality studies (Zhou et al. 2017; Ma et al. 2019), water treatment can significantly reduce pollutions from this source, and thus exerts a positive effect on improving surface water quality. As the reviewer said, different cities may decide to spend more or less money and resources to address pollution, and the decision will finally be reflected by surface water quality conditions. In this paper, we used in-situ water quality measurements to assess pollution induced water scarcity WSpol, the result of which has included the effect of investments of water treatment and other factors (e.g. anthropogenic pollution loadings and self-cleansing ability of water bodies) in different cities. We recognized this in the revised manuscript:

“Notwithstanding a recent trend in improved water quality, our results show that quality still presents a great issue for achieving safe water supply in China.” (Page 6, lines 209-211)

Quantitative contributions of investments to improved water quality and the regional difference have been discussed in detail in a recent study (Ma et al., 2020). We have cited this reference in the revised manuscript:

“Chinese state and local governments have invested thousands of billions of RMB Yuan to enhance wastewater treatment and control pollution sources dedicated to environmental restoration in the recent decade³⁷. Contributions of the investments on the improved surface water quality (characterized mainly by COD and NH₄⁺-N concentrations) and the regional difference were quantified in a recent study⁴⁶” (Page 6, lines 205-209)

2) The authors mention that groundwater cannot be included because of lack of data. This is a serious omission. Are there reasons not to consider using GRACE data for the more aggregated

geographical scales? If the author's have a good reason not to consider the groundwater data, I believe that they must still discuss the role of groundwater more than they do currently.

Author responses:

In quantity based water scarcity assessment, WSqua is assessed by the ratio of water uses to water availability deducted with EFR by the above Equation c3. As the water availability including both surface and ground water resources availability, the separation of surface and ground water uses is not necessary and it will not change WSqua. Because groundwater quality is usually better than surface water quality, the differentiation of groundwater and surface water qualities helps obtain more accurate pollution induced water scarcity WSpol. This requires high resolution groundwater quality data at the national level, which is not available, as stated in the manuscript. Using surface water quality to roughly represent groundwater quality will generally lead to over-estimation of WSpol in regions where surface water quality is inadequate for specific uses and a significant proportion of groundwater is exploited. To the best of our knowledge, GRACE data is for groundwater quantity assessment, and is thus not helpful in our quality based water scarcity assessment.

We have illustrated this clearly and recognized the limitation in the revised manuscript:

“It has to be recognized that we do not differentiate groundwater withdrawal and quality from surface water in this research, because of a lack of nationwide groundwater quality data. Quality from both surface water and groundwater for sectoral water uses is represented by in-situ surface water quality. This will not have any impact on WSqua. However, in regions where surface water quality is inadequate for specific uses and a significant proportion of groundwater is exploited, WSpol is expected lower in reality, as groundwater quality is often superior to surface water” (Pages 7, lines 261-267)

For the detailed explanations on the groundwater impact on WSpol, please refer to the response to the second-from-bottom comment from Reviewer #1.

3) The author's shorthand WSq, WSp, and WSc, are not intuitive and are difficult to keep track of.

Author responses:

We have added illustrations and changed those abbreviations to make them be easier tracked:

“Quantity based water scarcity (referred to as WSqua, see Methods), quality based water scarcity (also pollution induced water scarcity, abbreviated as WSpol) and water scarcity based on the combined effect of both quality and quantity (referred to as WScom) in present-day China exhibit geographical variations at the grid cell-level” (Page 3, lines 79-81)

4) Fig 3 is truly amazing and informative. It would be good for the authors to discuss this in more depth.

Author responses:

As the reviewer suggested, we have added more in-depth discussion on the impact of geographic scale on seasonal water scarcity based on Fig. 3 in the revised manuscript:

“Fig.3 also shows that seasonal water scarcity is sensitive to the geographic scale. In the most water scarce spring, basins under WScom comprise 70%, 69.7% and 71.3% of the total numbers of basins at the first-order, second-order and third-order basin levels, respectively; 39.3% grid-cells suffer water scarcity. The season with the least number of geographic units under WScom is summer or autumn, depending on the geographic scale.” (Page 4, lines 121-125)

5) The methodological approach using dilution is not explained to the reader, and should be.

Author responses:

The methodological approach using dilution to calculate pollution induced water scarcity $WSpol$ is now explained in detail in the methodology section in the revised manuscript:

“Adapting from van Vliet *et al.*³⁴, a dilution approach is applied to translate inadequate water quality into extra water quantity required so that it can be compared with water availability for water scarcity assessment. Quality based water scarcity $WSpol$ is calculated as the ratio of water required for dilution to obtain adequate quality for water uses to water availability:

$$WSpol = \frac{\sum_i dq_i}{Q - EFR} \quad (2)$$

with

$$dq_i = \max(dq_{i,j})$$
$$dq_{i,j} = \begin{cases} 0, & C_j \leq C_{max_{i,j}} \\ D_i \left(\frac{C_j}{C_{max_{i,j}}} - 1 \right), & C_j > C_{max_{i,j}} \end{cases} \quad (3)$$

where dq_i is extra water required for dilution to obtain acceptable quality for water use sector i , $dq_{i,j}$ indicates the amount of dilution water for sector i based on water quality parameter j , C_j is actual water quality level of parameter j , and $C_{max_{i,j}}$ is the maximum water quality threshold based on parameter j for water use sector i . Equation (3) is slightly different from its original form³⁴. Instead of diluting all the available water to obtain water quantity needed for a sectoral water use, we estimate the extra amount of water by diluting the volume of the sectoral water withdrawal. Quality constraint for EFR is not considered in this study.” (Pages 8, lines 299-311)

6) The policy implications are not clear. For example, would the authors recommend more investment in interregional water transfers? E.g. China’s South-North water transfer project? How does the discussion of variation across multiple geographic and temporal scales influence the larger takeaways? Influence policy recommendations?

Author responses:

As the reviewer suggested, we have added a few sentences on policy implications in the revised manuscript. As our study shows high regional inequality of water scarcity in China, interregional water transfers are recommended in combination with locally-specific water management policies:

“High regional inequality of water scarcity in China urges locally-specific policies for water demand management. In the meantime, inter-basin water transfers can directly mitigate water scarcity by supplying extra water sources to water scarce basins. Water scarcity assessment with a finer resolution corresponds to a greater inequality, indicating that water allocations within and between basins will be effective in reducing water scarcity inequality. However, the impact of water transfers on both source basins (e.g. social and eco-environmental impacts) and receiving basins (e.g. non-native species invasions and spreading of chemical/biological contaminants) needs to be carefully analyzed to minimize negative consequences.” (Page 7-8, lines 275-282)

As we have shown in this study, water scarcity is sensitive to geographic and temporal scales. We recommend a comprehensive water scarcity analysis at multiple geographic and temporal

scales, which provide lower and upper bounds of water scarcity levels due to uncertainty in practical water uses and management. We highlighted that:

“Our estimates of water scarcity intervals by different spatiotemporal scales also provide insights in the extent to which a corresponding measure (i.e. water resources allocations between seasons or sites) can be useful in coping with water scarcity. For instance, in basins where people suffer water scarcity in all the months, seasonal flow regulation is not effective in reducing water scarcity. Because human interventions (e.g. man-made reservoirs and human water uses) as well as legal and institutional arrangements may all result in changes in water scarcity, more detailed information from improved monitoring and management reporting needs to be incorporated in future water scarcity assessment to reduce uncertainty.” (Page 7, lines 248-255)

“Water scarcity assessment with a finer resolution corresponds to a greater inequality, indicating that water allocations within and between basins will be effective in reducing water scarcity inequality.” (Page 7-8, lines 275-279)

We also mentioned in the Discussion of the manuscript:

“The metric WSpol used in this study for measuring quality induced water scarcity, i.e. the ratio of water required for dilution to obtain adequate quality for water uses to water availability, is useful in guiding policy-making to identify basins where water supplies are mostly threatened by degraded water quality. At the third-order basin level (at which aquatic environment restorations are often planned and managed in China), prominent examples of water quality inferior to level IV (according to the “Environmental Quality Standard for Surface Water in China”) include basins in Hai River, Yellow River and Liao River basins (Supplementary Fig. S11). Given that the investment budget into aquatic environment restoration is limited each year, our results suggest that the need for water quality improvement in the Hai River sub-basins is the most urgent due to the severe scarcity ($WSpol > 2$).” (Pages 6, lines 212-220)

Minor points:

-The writing is not as clear as it could be and should be copyedited for grammar and style

Author responses:

All the authors (including a native English speaker) have read through the manuscript and carefully revised it to ensure correct grammar and consistent style of the manuscript.

-The headline range of the number to be affected (.7-1.36 billion) is too large to be informative -- more attention to multiple-scale approach would be better

Author responses:

We have changed the headline in the Results section to “Population under water scarcity at multiple scales”. (Page 6, line 159)

In addition, we have revised the abstract: “Over half of the population are affected by water scarcity” (Page 2, lines 30-31) we also highlighted the necessity of including multiple-scale analysis in water scarcity: “We highlight the necessity of incorporating water scarcity assessment at multiple temporal and geographic scales.” (Page 2, lines 26-27)

-The argument for analysis at multiple temporal and geographic scales is important, the authors could bring this to the forefront

Author responses:

As the reviewer suggested, we have now highlighted this in the revised manuscript. In the abstract, we state: “We highlight the necessity of incorporating water scarcity assessment at multiple temporal and geographic scales.” (Page 2, lines 26-27)

In addition, we have added a few sentence in Discussion to highlight the argument for implementing water scarcity assessment at multiple temporal and geographic scales:

“We argue here that it is essential to conduct water scarcity analyses at different temporal and geographic scales, unless sufficient evidence supports a specific scale. Analyses at different geographic and temporal scales provide lower and upper bounds of water scarcity levels due to uncertainty in practical water uses and management. Our estimates of water scarcity intervals at various spatiotemporal scales also provide insights in the extent to which a corresponding measure (i.e. water resources allocations between seasons or sites) can be useful in coping with water scarcity. For instance, in basins where people suffer water scarcity in all the months, seasonal flow regulation will not be effective in reducing water scarcity.” (Page 7, lines 245-252)

Reviewer #3 (Remarks to the Author)

This study is useful since authors have highlighted the importance of quality of water although quality and quantity are equally important. The work involves considerable data compilation from different sources as enumerated. The spatial presentation of results nationwide is highly useful for policymakers. The use of simple statistics is highly appreciated. The broad generalization arrived at spatially sounds a bit simplistic for many reasons. I would have liked to see an atlas of the agricultural carrying capacity of water on the basis of water quality; industrial carrying capacity on the basis of water quality and also human and animal habitat carrying capacity on the basis of quality of water. Then how quality coupled with the quantity of water can be used to modify the carrying capacity.

Author responses:

We would like to sincerely thank the reviewer for the overall encouraging feedback to our work. We agree with the reviewer that an atlas of carrying capacities of water including water quality aspect is useful in examining the impact of water quality on local life and productions. As suggested, we have added carrying capacity maps and relevant explanations in the revised manuscript, as detailed below.

In Supplementary Information, we have defined quantity based and quality included water carrying capacities. Water carrying capacities for different sectors are discussed in detail. We have evaluated the water carrying capacity in terms of a multiplier of present-day population or economic scale that can be sustained by water endowment and quality conditions. The equations for both quantity based and quality included water carrying capacities have been provided. The physical meaning of water carrying capacities is discussed, and water carrying capacities at the first-order basin level in China is presented:

“A water carrying capacity C_{qua} or C_{com} larger than 1 would mean that the water resources are able to sustainably support the current population and economic scale; this would correspond to low water scarcity, with a WS_{qua} or WS_{com} of less than 1. In contrast, a water carrying capacity C_{qua} or C_{com} below 1 would indicate the insufficient carrying capacity of water resources to support the present-day population and economic scale; this would correspond to a moderate-to-severe water scarcity issue, with a WS_{qua} or WS_{com} of over 1. Thus understood, the value of C_{qua} or C_{com} is the reciprocal of the value of WS_{qua} or WS_{com} . Fig. S12 shows the water carrying capacity in terms of a multiplier of present-day population and economic scale at the first-order basin level on an annual basis. In this case, as the agricultural water use constitutes a significant portion of the total water use, the carrying capacity for agriculture is the lowest in all the sectors. In the most-water-scarce Haihe River basin, the water carrying capacity is not able to support each of the sectoral water uses alone. Carrying capacity C_{com} in all six North China basins and in the Southeast River basin is less than 1, indicating an insufficient capacity of water resources for sustaining the population and economic activities in these basins (Supplementary Fig. S12h). This is consistent with the water scarcity condition of these basins, as shown in Fig. 2(c). Supplementary Fig. S12 also shows that inadequate water quality reduces the carrying capacity of water resources.” (Pages 7-8, lines 149-165 in Supplementary Information)

Water carrying capacity is discussed in the revised manuscript due to limited space:

“The impact of water quality on domestic and production water uses can also be analyzed using the concept of water carrying capacity, which refers to maximum population or economic scale

that can be sustainably supported by available water resources. The water carrying capacity, in terms of a multiplier of present-day population or economic scale, is shown in Supplementary Information, and similar conclusions can be derived as those from WSqua and WScorn assessments (see Supplementary Text and Fig.S12 for more details).” (Page 6, lines 220-226)

Another aspect is water quality study need to be connected with the water usage pattern which presents different scenarios. Water for cooking and drinking have different quality criteria than washing and flushing the toilet. You can also look at such usage needs in industry and agriculture. It will be useful if you highlight with the relevant database the issue of carrying capacity and usage pattern

Author responses:

We agree with the reviewer that sectoral water uses categorized in higher resolutions will improve the accuracy of quality based water scarcity assessment. As the reviewer suggested, different quality requirements for drinking and toilet flushing water uses (both in the domestic water use category in this study) may affect water scarcity assessment result. However, in China, domestic water is mostly supplied to households by water supply networks, without differentiating drinking and other domestic uses. Though grey or recycled water has been used in some cities, mainly for public water uses such as watering municipal parks and green spaces, the share of recycled water is very low and is almost negligible. The extra costs for building and maintaining dual water distribution systems to households and residents’ unwillingness to use recycled water are large barriers to extensive use of recycled water in China. Therefore, in practice, quality requirements for various domestic water uses, including drinking, washing and toilet flushing, do not present any difference due to current water supply systems. It is also true that specific industries and irrigated crops may have specific water quality requirements. However, because data for both water uses and quality requirements with high-resolution categories are not readily available, it is difficult to implement quality included water scarcity assessment with more specified water use categories (rather than 4 categories of water uses, i.e. agricultural, industrial, domestic and ecological water uses), in this nationwide study of such a large spatial scale. We have explained this in the Supplementary Information:

“Water quality requirements C_{max} in Equation (3) are defined for agricultural, industrial, domestic and eco-environmental compensation uses. Though quality requirements may present difference for water uses at a higher resolution in one sector, categorization of sectoral water uses for quality-induced water scarcity assessment in this nationwide study is based on data availability and best practices in China. For instance, drinking and toilet flushing water uses (both in the domestic water use sector) may differ in water quality requirements. However, the extra costs for building and maintaining dual water distribution systems to households and residents’ unwillingness to use recycled water are large barriers to extensive use of recycled water for domestic use. Domestic water is mostly supplied to households by water supply networks, without differentiating drinking and other domestic uses. Therefore, in practice, quality requirements for various domestic water uses, including drinking, washing and toilet flushing, do not present any difference due to current water supply systems. Specific industries and irrigated crops may also have different water quality requirements. However, because data for both water uses and quality requirements with sectors of more detailed categories are not readily available, it is difficult to

implement quality included water scarcity assessment with more specified water use categories.”
(Page 4, lines 26-39 in Supplementary Information)

References

- [1] Hoekstra, A., Mekonnen, M., Chapagain, K., Mathews, R. & Richter, B. Global monthly water scarcity: Blue water footprints versus blue water availability. *PLoS One* **7**, e32688 (2012).
- [2] Mekonnen, M. & Hoekstra, A. Four billion people facing severe water scarcity. *Sci. Adv.* **2016**: **2**, e1500323 (2016).
- [3] Wada, Y., van Beek, L. & Bierkens, M. Modelling global water stress of the recent past: on the relative importance of trends in water demand and climate variability *Hydrol. Earth Syst. Sci.* **15**, 3785–808 (2011).
- [4] Pastor A.V. et al. Accounting for environmental flow requirements in global water assessments. *Hydrol. Earth Syst. Sci.* **18**, 5041–5059, 2014
- [5] Liu, J. et al. Water scarcity assessments in the past, present and future. *Earth's Future* **5**, 545–559 (2017)
- [6] Hanasaki, N. *et al.* Quantitative investigation of the thresholds for two conventional water scarcity indicators using a state-of-the-art global hydrological model with human activities. *Water Resources Research* **54**, 8279-8294. (2018)
- [7] Ministry of Ecology and Environmental Protection of China, Environmental quality standards for surface water (GB 3838-2002) (2002).
- [8] Zhou, Y. et al. Improving water quality in China: Environmental investment pays dividends. *Water Res.* **118**, 152–159 (2017).
- [9] Ma T. et al. China's improving inland surface water quality since 2003. *Sci. Adv.* (In press, 2020)
- [10] Ayers, R. & Westcot, D. Water quality for agriculture. Food and Agriculture Organization of the United Nations (FAO), Rome (1985). (Available at <http://www.fao.org/3/T0234E/T0234E01.htm>)

REVIEWERS' COMMENTS:

Reviewer #1 (Remarks to the Author):

Thanks for the revised version.

I have reviewed the revised manuscript and responses to the reviewer comments, which have been addressed well.

It's a very interesting study and clearly demonstrates the importance of including water quality and impact of spatial/temporal scale issues for water scarcity assessment. I would recommend having the revised manuscript accepted for publication.

Reviewer #2 (Remarks to the Author):

The authors have carefully responded to the reviewers' suggestions for improving the manuscript. Given the importance of the research questions, extensive data collection efforts, and the thoughtful revisions, I recommend publication.

Responses to reviewers

NCOMMS-19-29361A

Title: Pollution exacerbates China's water scarcity and its regional inequality

Reviewer #1 (Remarks to the Author)

Thanks for the revised version.

I have reviewed the revised manuscript and responses to the reviewer comments, which have been addressed well. It's a very interesting study and clearly demonstrates the importance of including water quality and impact of spatial/temporal scale issues for water scarcity assessment. I would recommend having the revised manuscript accepted for publication.

Author responses:

Thank you for your helpful comments and suggestions for improving our manuscript.

Reviewer #2 (Remarks to the Author)

The authors have carefully responded to the reviewers' suggestions for improving the manuscript. Given the importance of the research questions, extensive data collection efforts, and the thoughtful revisions, I recommend publication.

Author responses:

Thank you for your helpful comments and suggestions for improving our manuscript.